**Data Availability Statement:** All relevant data are within the manuscript and its Supporting Information files.

# A polymorphism in the cachexia-associated gene *INHBA* predicts efficacy of regorafenib in patients with refractory metastatic colorectal cancer

Yuji Miyamoto[1], Marta Schirripa[1], Mitsukuni Suenaga[1], Shu Cao[2], Wu Zhang[1], Satoshi Okazaki[1], Martin D. Berger[1], Satoshi Matsusaka[1], Dongyun Yang[2], Yan Ning[1], Hideo Baba[3], Fotios Loupakis[4], Sara Lonardi[4], Filippo Pietrantonio[5], Beatrice Borelli[6], Chiara Cremolini[6], Toshiharu Yamaguchi[7], Heinz-Josef Lenz[1]*

1 Division of Medical Oncology, Norris Comprehensive Cancer Center, Keck School of Medicine, University of Southern California, Los Angeles, CA, United States of America, 2 Department of Preventive Medicine, Norris Comprehensive Cancer Center, Keck School of Medicine, University of Southern California, Los Angeles, CA, United States of America, 3 Department of Gastroenterological Surgery, Graduate School of Medical Sciences, Kumamoto University, Kumamoto, Japan, 4 Unit of Medical Oncology 1, Department of Clinical and Experimental Oncology, Istituto Oncologico Veneto, IRCCS, Padua, Italy, 5 Medical Oncology Department, Fondazione IRCCS Istituto Nazionale dei Tumori, Milan, Italy, 6 Polo Oncologico, Azienda Ospedaliero-Universitaria Pisana, Pisa, Italy, 7 Department of Gastroenterological Surgery, Cancer Institute Hospital, Japanese Foundation for Cancer Research, Tokyo, Japan

* lenz@usc.edu

## Abstract

Activin/myostatin signaling has a critical role not only in cachexia but also in tumor angiogenesis. Cachexia is a frequent complication among patients with advanced cancer and heavily pretreated patients. We aimed to evaluate the prognostic significance of cachexia-associated genetic variants in refractory metastatic colorectal cancer (mCRC) patients treated with regorafenib. Associations between twelve single nucleotide polymorphisms in 8 genes (*INHBA*, *MSTN*, *ALK4*, *TGFBR1*, *ALK7*, *ACVR2B*, *SMAD2*, *FOXO3*) and clinical outcome were evaluated in mCRC patients of three cohorts: a discovery cohort of 150 patients receiving regorafenib, a validation cohort of 80 patients receiving regorafenib and a control cohort of 128 receiving TAS-102. In the discovery cohort, patients with any G variant in *FOXO3* rs12212067 had a significantly lower response rate ($P = 0.031$) and overall survival (OS) than those with a T/T in univariate analysis (4.5 vs. 7.6 months, hazard ratio [HR] = 1.63, 95% confidence interval [CI] = 1.09–2.46, $P = 0.012$). Among female patients, those with any G variant in *INHBA* rs2237432 had a significantly longer OS than those with an A/A in both univariate (7.6 vs. 4.3 months, HR = 0.57, 95%CI = 0.34–0.95, $P = 0.021$) and multivariable (HR = 0.53, 95%CI = 0.29–0.94, adjusted $P = 0.031$) analysis. This association was confirmed in female patients of the validation cohort, though without statistical significance ($P = 0.059$). Conversely, female patients with any G allele in the control group receiving TAS-102 did not show a longer OS. This was the first study evaluating the associations between polymorphisms in cachexia-associated genes and outcomes in refractory mCRC

**Funding:** This work was supported by the National Institute of Health (P30CA014089); the Gloria Borges Wunderglo Project; the Dhont Family Foundation; the Daniel Butler Research Fund; and the Call to Cure Research Fund. YM received a grant from Japan Society for the Promotion of Science (S2606). MS is the recipient of Takashi Tsuruo Memorial Fund. MDB received a grant from the Werner and Hedy Berger-Janser Foundation for cancer research and the Swiss Cancer League (BIL KLS-3334-02-2014). The funders had no role in study design, data collection and analysis, decision to publish, or preparation of the manuscript.

**Competing interests:** The authors have read the journal's policy and the authors have the following competing interests: HJ Lenz has received honoraria from Merck Serono, Roche, Celgene, Bayer, and Boehringer Ingelheim. HB received honoraria from Chugai Pharma, Bayer, Taiho Pharmaceutical and Merck Serono. There are no patents, products in development or marketed products to declare. This does not alter our adherence to PLOS ONE policies on sharing data and materials.

patients treated with regorafenib. Further studies should be conducted to confirm these associations.

## Introduction

Regorafenib is a small molecule multikinase inhibitor that blocks protein kinases involved in tumor angiogenesis, oncogenesis and the tumor microenvironment [1]. The benefit of regorafenib on overall survival (OS) in patients with metastatic colorectal cancer (mCRC) was demonstrated in two phase III randomized controlled trials, the CORRECT [2] and CONCUR [3] trials. Therefore, regorafenib is now established as an additional line of therapy for patients with mCRC refractory to previous chemotherapy as well as for best supportive care [4, 5]. Several investigators have attempted to identify molecular markers that predict the activity of regorafenib for the individualized treatment of patients with mCRC. For example, expression levels of biomarkers such as *VEGF* and *CCL5* [6] or plasma circulating cell-free DNA [7] may represent potential predictive biomarkers of regorafenib treatment, although these results have not been sufficiently validated.

Cancer cachexia is defined as an ongoing loss of skeletal muscle mass and is a more common complication in heavily pretreated cancer patients [8], leading to progressive impairment of physical function and quality of life as well as resistance to chemotherapy or radiotherapy [9, 10]. Skeletal muscle mass is dynamically regulated by various extracellular signals, which activate distinct intracellular signaling processes [11]. In particular, *INHBA* and *MSTN* are potent negative regulators of muscle mass [12]. The binding of *INHBA* and *MSTN* to membrane receptors (*ACVR1B*, *C*, and *ACVR2B*) leads to the activation of *SMAD*-mediated signal transduction, promoting muscle protein degradation [13]. *INHBA* or *MSTN* expression is associated with several types of human cancers, and CRC patients with high INHBA expression showed poorer OS than those with low INHBA expression [14]. In addition, accumulating evidence suggests that activin/myostatin signaling, like other members of the *TGF-beta* superfamily, can regulate angiogenesis. *MSTN* blockade reduced the tumor expression of genes involved in angiogenesis (e.g. *VEGF-A*, *HIF-1α*) [15]. Similarly, *INHBA* demonstrated both pro- [16] and anti-angiogenic [17] properties in different systems. Recently, we reported that germline variants within the cancer cachexia pathway are associated with outcome in mCRC patients treated with bevacizumab-based chemotherapy [18].

Based on the clinical importance of cachexia signaling being potentially involved in angiogenesis, we evaluated the prognostic and predictive significance of cachexia-associated genetic variants in refractory mCRC patients treated with regorafenib chemotherapy. A previous report indicated that gender differences may influence skeletal muscle changes after chemotherapy [19]. We therefore determined whether such associations were influenced by gender.

## Materials and methods

### Study design and patients

This study was a retrospective exploratory study in three independent cohorts of patients with refractory mCRC: a discovery cohort of 150 patients receiving regorafenib at Azienda Ospedaliero-Universitaria Pisana, Istituto Oncologico Veneto (Padova, Italy); a validation cohort of 80 patients receiving regorafenib at the Cancer Institute Hospital of the Japanese Foundation for Cancer Research (Japan); and a control cohort of 128 patients receiving TAS-102 at Azienda Ospedaliero-Universitaria Pisana, Istituto Oncologico Veneto (Padova, Italy) and

Istituto Nazionale Tumori (Milano, Italy). Patients with histologically verified colorectal adenocarcinoma, measurable metastatic disease according to Response Evaluation Criteria in Solid Tumors (RECIST) v1.1, and a history of previous standard chemotherapy with 5-FU, L-OHP, CPT-11, bevacizumab, and cetuximab or panitumumab were eligible. Patients received regorafenib 160 mg per body once daily from days 1–21, every 4 weeks, or TAS-102 35 mg per m$^2$ twice daily on days 1–5 and 8–12, every 4 weeks. Treatment was administered until disease progression, intolerable toxicities, or patient withdrawal occurred. All patients provided written informed consent, including consent for all medical record which were fully anonymized before we assessed, blood or tumor tissue to be used to explore relevant molecular parameters. This study was conducted according to the REporting recommendations for tumor MARKer prognostic studies (REMARK) [20]. The tissue analysis protocol was approved by the University of Southern California (USC) Institutional Review Board of Medical Sciences and conducted at the USC/Norris Comprehensive Cancer Center in accordance with the Declaration of Helsinki and Good Clinical Practice guidelines.

## Selection of candidate single-nucleotide polymorphisms

The 12 candidate single nucleotide polymorphisms (SNPs) in the cachexia pathway examined in this study were *INHBA*, *MSTN*, *ALK4*, *TGFBR1*, *ALK7*, *ACVR2B*, *SMAD2*, and *FOXO3*, which were selected using one of the following criteria: i) SNP with potential biological significance based on the published literature or F-SNP database* (http://compbio.cs.queensu.ca/F-SNP/); or ii) minor allele frequency ≥10% in both white and East Asians in the Ensembl Genome Browser. The characteristics of the selected polymorphisms are shown in S1 Table.

## DNA extraction and genotyping

Genomic DNA was extracted from patients' peripheral blood using a QIAmp Kit (Qiagen, Valencia, CA, USA) according to the manufacturer's protocol. The candidate SNPs were examined by PCR-based direct DNA sequence analysis using an ABI 3100A Capillary Genetic Analyzer and Sequencing Scanner v1.0 (Applied Biosystems, Foster City, CA, USA). The primers for amplification of extracted DNA are listed in S1 Table. For quality control purposes, 10% of samples were randomly selected and analyzed by direct DNA sequencing for each SNP. The genotype concordance rate was found to be ≥99%. The investigators analyzing SNPs were blinded to the clinical data.

## Statistical analysis

The primary endpoint in this study was progression-free survival (PFS), and the secondary endpoints were OS and disease control rate (DCR). PFS was defined as the period between the date of starting treatment and the date of confirmed disease progression or death. OS was calculated from the date of starting treatment until the date of death from any cause. If the event was not observed by the last follow up date, the patient was recorded as censored. In patients lost to follow-up, data were censored at the date of last follow up. According to RECIST v1.1, DCR was defined as the proportion of patients who achieved stable disease (SD), partial response (PR), or progressive disease (PD). Chi-square tests were used to examine the difference in baseline patient characteristics between the three cohorts. Allelic distribution of polymorphisms was tested for deviation from the Hardy–Weinberg equilibrium using the exact test. Linkage disequilibrium among SNPs was evaluated using *D'* and *r$^2$* values and haplotype frequencies of genes were inferred using HaploView version 4.2 (http://www.broad.mit.edu/mpg/haploview). High linkage disequilibrium was defined as r2 > 0.7. Fisher's exact test was applied to examine the associations between SNPs and DCR. Associations between candidate

SNPs and PFS or OS were analyzed by the Kaplan–Meier method and log-rank test in the univariable analysis and reevaluated using a Cox proportional hazards model and Wald test with predictive or prognostic baseline factors included. The baseline demographic and clinical characteristics statistically significantly associated with PFS and OS in multivariable analyses were included in the final models. We used codominant, dominant, or recessive genetic models where appropriate for the candidate SNPs, because the true modes were not yet established in the analyses. The minimum detectable hazard ratios of 1.61–1.82 corresponded to the minor allele frequency of 0.1–0.4 in the association between an SNP and PFS in the discovery cohort (n = 150, PFS events = 149), considering a dominant model and using a two-sided 0.05-level log-rank test with 80% power. In the validation cohort (n = 80, PFS events = 79), the power was 54% using the same model. All analyses were carried out with SAS 9.4 (SAS Institute, Cary, NC, USA). All tests were two-sided at a significance level of 0.050. *P*-act method, a modified multiple testing method, was applied for adjusting the *P* values for all SNPs when the linkage disequilibrium between candidate SNPs and different modes of inheritance was considered.

## Results

### Baseline characteristics

The baseline characteristics of enrolled cohorts are summarized in Table 1. Gender, performance status, adjuvant treatment history, and the number of prior chemotherapy regimens were distributed differently between the cohorts. The median PFS, OS, and follow-up time were 2.1, 6.0, and 36.4 months in the discovery cohort; 2.0, 8.0, and 15.3 months in the validation cohort; and 2.0, 5.4, and 5.3 months in the control cohort. Genotyping was successful in at least 90% of cases for each polymorphism analyzed. The allelic frequencies for all SNPs were within the probability limits of the Hardy–Weinberg equilibrium (*P*>0.050). High linkage disequilibrium was observed between *ACVR2B* rs13072731 and *ACVR2B* rs2268753 in the discovery cohort, with *D'* = 0.98 and *r²* = 0.70. No other high linkage disequilibrium was observed between the SNPs found in each cohort.

### Associations between cachexia SNPs and outcome in the discovery and validation cohorts

Associations between candidate SNPs and clinical outcome were examined in the regorafenib discovery cohort. Patients with any G allele in *FOXO3* rs12212067 had significantly shorter PFS and OS and worse DCR than those with a T/T variant in univariate analysis (PFS: 1.8 vs. 2.1 months, hazard ratio [HR] 1.44, 95% confidence interval [CI] 0.98–2.12, *P* = 0.056; OS: 4.5 vs. 7.6 months, HR 1.63, 95% CI 1.09–2.46, *P* = 0.012; DCR: *P* = 0.031) (Table 2). However, in the validation cohort, patients with any G allele in *FOXO3* rs12212067 had longer PFS and OS than those with a T/T variant in univariate analysis (PFS: 2.0 vs. 2.5 months, HR 0.56, 95% CI 0.32–0.97, *P* = 0.027; OS: 7.6 vs. 15.3 months, HR 0.49, 95% CI 0.23–1.04, *P* = 0.054) (S2 Table). However, these effects were not significant in the multivariable model and after multiple testing.

### Associations between cachexia SNPs and outcome stratified by sex in the discovery and validation cohorts

Among female patients in the discovery cohort, patients with any G allele in *INHBA* rs2237432 showed a significantly longer OS than those with the A/A allele in both univariate (7.6 vs. 4.3 months, HR 0.57, 95% CI 0.34–0.95, *P* = 0.021) and multivariable analysis (HR 0.53, 95% CI

**Table 1. Baseline clinical characteristics of patients in the discovery, validation, and control cohorts.**

| Characteristics | Discovery cohort (n = 150) | | Validation cohort (n = 80) | | Control cohort (n = 128) | | P value [a] |
|---|---|---|---|---|---|---|---|
| | N | (%) | N | (%) | N | (%) | |
| Age (years) | | | | | | | 0.97 |
| ≤65 | 94 | (63) | 49 | (61) | 79 | (62) | |
| >65 | 56 | (37) | 31 | (39) | 49 | (38) | |
| Gender | | | | | | | 0.49 |
| Male | 81 | (54) | 38 | (48) | 61 | (48) | |
| Female | 69 | (46) | 42 | (53) | 67 | (52) | |
| Primary tumor site | | | | | | | 0.80 |
| Right | 49 | (33) | 23 | (29) | 39 | (30) | |
| Left | 99 | (66) | 57 | (71) | 85 | (66) | |
| Unknown [b] | | | | | 4 | (3) | |
| Primary tumor resected | | | | | | | 0.52 |
| Yes | 127 | (85) | 70 | (88) | 109 | (85) | |
| No | 23 | (15) | 10 | (12) | 13 | (10) | |
| Unknown [b] | | | | | 6 | (5) | |
| Adjuvant treatment | | | | | | | 0.087 |
| Yes | 37 | (25) | 27 | (34) | 47 | (37) | |
| No | 112 | (75) | 53 | (66) | 81 | (63) | |
| Unknown [b] | 1 | (0) | | | | | |
| Liver metastasis | | | | | | | 0.11 |
| Yes | 120 | (80) | 54 | (68) | 96 | (75) | |
| No | 30 | (20) | 26 | (33) | 32 | (25) | |
| Lung metastasis | | | | | | | 0.096 |
| Yes | 109 | (73) | 47 | (59) | 88 | (69) | |
| No | 41 | (27) | 33 | (41) | 40 | (31) | |
| LN metastasis | | | | | | | 0.47 |
| Yes | 75 | (50) | 41 | (51) | 56 | (44) | |
| No | 75 | (50) | 39 | (49) | 72 | (56) | |
| Peritoneal involvement | | | | | | | 0.67 |
| Yes | 43 | (29) | 20 | (25) | 31 | (24) | |
| No | 107 | (71) | 60 | (75) | 97 | (76) | |
| Number of metastases | | | | | | | 0.001 |
| 1 | 16 | (11) | 24 | (30) | 21 | (16) | |
| >1 | 134 | (89) | 56 | (70) | 107 | (84) | |
| Number of treatment regimens | | | | | | | < 0.001 |
| ≤ 3 | 108 | (72) | 72 | (90) | 80 | (63) | |
| > 3 | 42 | (28) | 8 | (10) | 48 | (38) | |
| Performance status | | | | | | | < 0.001 |
| ECOG 0 | 117 | (78) | 45 | (56) | 72 | (56) | |
| ECOG 1 | 33 | (22) | 35 | (44) | 56 | (44) | |
| RAS status | | | | | | | 0.41 |
| Wild | 52 | (35) | - | | 47 | (37) | |
| Mutant | 93 | (62) | - | | 68 | (53) | |
| Unknown [b] | 5 | (3) | - | | 13 | (10) | |

Abbreviations: LN, lymph node; ECOG, Eastern Cooperative Oncology Group.

[a] Based on the χ2 test.

[b] Not included in the test.

**Table 2. Association between cachexia-related gene polymorphism and clinical outcomes in the discovery cohort.**

| Genotype | N | Tumor response | | | Progression-free survival | | | | | Overall survival | | | | |
|---|---|---|---|---|---|---|---|---|---|---|---|---|---|---|
| | | PR+SD | PD | *P* value* | Median, months (95%CI) | HR (95%CI) † | *P* value* | HR (95%CI) ‡ | *P* value* | Median, months (95%CI) | HR (95%CI) † | *P* value* | HR (95%CI) ‡ | *P* value* |
| *INHBA* rs2237432 | | | | 0.91 | | | 0.92 | | 0.41 | | | 0.59 | | 0.87 |
| A/A | 78 | 27 (37%) | 46 (63%) | | 2.1 (1.8, 2.8) | 1 (Reference) | | 1 (Reference) | | 6.5 (4.5, 8.7) | 1 (Reference) | | 1 (Reference) | |
| A/G | 56 | 18 (33%) | 37 (67%) | | 2.1 (1.8, 2.3) | 0.94 (0.66, 1.34) | | 0.99 (0.68, 1.43) | | 5.6 (4.3, 7.6) | 0.91 (0.63, 1.31) | | 1.00 (0.68, 1.48) | |
| G/G | 16 | 5 (33%) | 10 (67%) | | 2.1 (1.7, 4.0) | 1.02 (0.59, 1.74) | | 1.44 (0.82, 2.54) | | 9.5 (2.2, 13.9) | 0.75 (0.42, 1.34) | | 1.17 (0.64, 2.12) | |
| *INHBA* rs17776182 | | | | 1.00 | | | 0.78 | | 0.68 | | | 0.21 | | 0.47 |
| G/G | 122 | 41 (35%) | 76 (65%) | | 2.1 (1.9, 2.3) | 1 (Reference) | | 1 (Reference) | | 5.9 (4.5, 7.8) | 1 (Reference) | | 1 (Reference) | |
| G/A [a] | 25 | 8 (35%) | 15 (65%) | | 1.9 (1.8, 3.6) | 0.94 (0.62, 1.43) | | 1.09 (0.71, 1.69) | | 8.7 (5.5, 12.4) | 0.77 (0.50, 1.18) | | 0.85 (0.54, 1.33) | |
| A/A [a] | 3 | 1 (33%) | 2 (67%) | | | | | | | | | | | |
| *MSTN* rs7570532 | | | | 0.46 | | | 0.67 | | 0.73 | | | 0.96 | | 0.98 |
| A/A | 89 | 33 (39%) | 51 (61%) | | 2.3 (1.9, 3.0) | 1 (Reference) | | 1 (Reference) | | 7.8 (5.7, 9.4) | 1 (Reference) | | 1 (Reference) | |
| A/G [a] | 54 | 15 (29%) | 37 (71%) | | 1.9 (1.8, 2.1) | 1.07 (0.77, 1.49) | | 1.06 (0.76, 1.48) | | 5.3 (3.6, 7.6) | 1.01 (0.71, 1.42) | | 1.00 (0.70, 1.41) | |
| G/G [a] | 7 | 2 (29%) | 5 (71%) | | | | | | | | | | | |
| *ALK4* rs2854464 | | | | 0.90 | | | 0.95 | | 0.77 | | | 0.48 | | 0.17 |
| A/A | 78 | 25 (34%) | 49 (66%) | | 2.1 (1.8, 2.8) | 1 (Reference) | | 1 (Reference) | | 6.0 (5.0, 8.9) | 1 (Reference) | | 1 (Reference) | |
| A/G | 56 | 18 (34%) | 35 (66%) | | 2.1 (1.8, 2.3) | 1.05 (0.74, 1.49) | | 1.13 (0.79, 1.61) | | 6.3 (4.4, 8.0) | 1.09 (0.76, 1.57) | | 1.14 (0.78, 1.66) | |
| G/G | 15 | 6 (40%) | 9 (60%) | | 2.2 (1.4, 3.8) | 1.06 (0.61, 1.85) | | 1.14 (0.65, 2.02) | | 5.7 (2.0, 10.0) | 1.41 (0.80, 2.47) | | 1.74 (0.97, 3.10) | |
| *TGFBR1* rs10760673 | | | | 0.57 | | | 0.60 | | 0.053 | | | 0.72 | | 0.28 |
| G/G | 92 | 32 (37%) | 54 (63%) | | 2.0 (1.8, 2.8) | 1 (Reference) | | 1 (Reference) | | 7.7 (5.4, 9.1) | 1 (Reference) | | 1 (Reference) | |
| G/A [a] | 48 | 17 (35%) | 31 (65%) | | 2.1 (1.9, 2.4) | 0.92 (0.65, 1.29) | | 0.70 (0.49, 1.00) | | 5.6 (4.4, 7.8) | 1.06 (0.75, 1.51) | | 0.82 (0.57, 1.18) | |
| A/A [a] | 8 | 1 (14%) | 6 (86%) | | | | | | | | | | | |
| *ALK7* rs13010956 | | | | 0.94 | | | 0.62 | | 0.52 | | | 0.95 | | 0.68 |
| T/T | 45 | 15 (34%) | 29 (66%) | | 1.9 (1.8, 2.3) | 1 (Reference) | | 1 (Reference) | | 5.4 (3.5, 7.9) | 1 (Reference) | | 1 (Reference) | |
| T/C | 80 | 26 (34%) | 51 (66%) | | 2.1 (1.9, 2.3) | 0.84 (0.58, 1.22) | | 0.83 (0.57, 1.20) | | 6.5 (5.5, 8.9) | 0.99 (0.67, 1.46) | | 0.93 (0.62, 1.38) | |
| C/C | 24 | 8 (38%) | 13 (62%) | | 1.9 (1.8, 4.1) | 0.84 (0.51, 1.39) | | 1.02 (0.60, 1.71) | | 7.4 (3.5, 9.7) | 1.07 (0.64, 1.79) | | 1.16 (0.68, 1.97) | |
| *ACVR2B* rs13072731 | | | | 0.30 | | | 0.89 | | 0.64 | | | 0.49 | | 0.33 |

(*Continued*)

**Table 2.** (Continued)

| Genotype | N | Tumor response | | | Progression-free survival | | | | | Overall survival | | | | |
|---|---|---|---|---|---|---|---|---|---|---|---|---|---|---|
| | | PR+SD | PD | *P* value* | Median, months (95%CI) | HR (95%CI) † | *P* value* | HR (95%CI) ‡ | *P* value* | Median, months (95%CI) | HR (95%CI) † | *P* value* | HR (95%CI) ‡ | *P* value* |
| C/C | 52 | 20 (40%) | 30 (60%) | | 2.1 (1.8, 3.1) | 1 (Reference) | | 1 (Reference) | | 6.0 (4.4, 10.5) | 1 (Reference) | | 1 (Reference) | |
| C/A | 73 | 25 (37%) | 43 (63%) | | 2.2 (1.9, 2.8) | 1.03 (0.72, 1.47) | | 1.12 (0.78, 1.62) | | 6.4 (4.7, 8.0) | 1.24 (0.85, 1.81) | | 1.35 (0.91, 1.99) | |
| A/A | 23 | 5 (22%) | 18 (78%) | | 1.8 (1.8, 2.1) | 0.92 (0.55, 1.55) | | 0.89 (0.52, 1.53) | | 7.6 (2.7, 9.7) | 1.05 (0.63, 1.76) | | 1.15 (0.68, 1.93) | |
| *ACVR2B* rs2268753 | | | | 0.56 | | | 0.80 | | 0.70 | | | 0.85 | | 0.67 |
| T/T | 38 | 15 (41%) | 22 (59%) | | 2.1 (1.8, 3.1) | 1 (Reference) | | 1 (Reference) | | 6.2 (4.4, 10.1) | 1 (Reference) | | 1 (Reference) | |
| T/C | 81 | 27 (35%) | 50 (65%) | | 2.2 (1.9, 2.7) | 0.88 (0.60, 1.30) | | 0.96 (0.65, 1.43) | | 6.0 (4.7, 8.0) | 1.12 (0.74, 1.69) | | 1.21 (0.80, 1.84) | |
| C/C | 31 | 8 (28%) | 21 (72%) | | 2.0 (1.8, 2.3) | 0.89 (0.54, 1.46) | | 0.81 (0.49, 1.35) | | 7.6 (3.3, 9.1) | 1.13 (0.69, 1.85) | | 1.15 (0.70, 1.89) | |
| *SMAD2* rs1792671 | | | | 0.52 | | | 0.18 | | 0.44 | | | 0.15 | | 0.58 |
| G/G | 55 | 18 (34%) | 35 (66%) | | 2.1 (1.8, 2.5) | 1 (Reference) | | 1 (Reference) | | 7.6 (5.1, 9.4) | 1 (Reference) | | 1 (Reference) | |
| G/A | 64 | 24 (40%) | 36 (60%) | | 2.3 (1.8, 3.7) | 0.78 (0.54, 1.13) | | 0.80 (0.54, 1.16) | | 7.0 (4.5, 9.6) | 0.88 (0.60, 1.28) | | 0.93 (0.63, 1.37) | |
| A/A | 30 | 8 (28%) | 21 (72%) | | 1.9 (1.8, 2.2) | 1.12 (0.71, 1.76) | | 0.99 (0.62, 1.59) | | 4.9 (3.1, 8.0) | 1.36 (0.85, 2.16) | | 1.21 (0.75, 1.97) | |
| *SMAD2* rs1792689 | | | | 0.49 | | | 0.49 | | 0.80 | | | 0.58 | | 0.50 |
| C/C | 109 | 35 (34%) | 69 (66%) | | 2.0 (1.8, 2.3) | 1 (Reference) | | 1 (Reference) | | 6.5 (5.3, 8.7) | 1 (Reference) | | 1 (Reference) | |
| C/T [a] | 38 | 13 (36%) | 23 (64%) | | 2.2 (1.8, 3.4) | 0.88 (0.62, 1.27) | | 0.95 (0.66, 1.38) | | 5.9 (3.7, 8.9) | 1.11 (0.76, 1.61) | | 1.14 (0.78, 1.68) | |
| T/T [a] | 3 | 2 (67%) | 1 (33%) | | | | | | | | | | | |
| *FOXO3* rs12212067 | | | | **0.031** | | | 0.056 | | 0.19 | | | **0.012** | | 0.094 |
| T/T | 114 | 42 (38%) | 68 (62%) | | 2.1 (1.9, 2.5) | 1 (Reference) | | 1 (Reference) | | 7.6 (5.9, 9.0) | 1 (Reference) | | 1 (Reference) | |
| T/G [a] | 34 | 6 (19%) | 25 (81%) | | 1.8 (1.8, 2.3) | 1.44 (0.98, 2.12) | | 1.31 (0.87, 1.96) | | 4.5 (2.7, 5.5) | 1.63 (1.09, 2.46) | | 1.43 (0.94, 2.17) | |
| G/G [a] | 1 | 1 (100%) | 0 (0%) | | | | | | | | | | | |
| *FOXO3* rs4946935 | | | | 0.051 | | | 0.29 | | 0.26 | | | 0.30 | | 0.36 |
| G/G | 87 | 31 (38%) | 51 (62%) | | 2.1 (1.9, 2.4) | 1 (Reference) | | 1 (Reference) | | 6.5 (5.5, 9.1) | 1 (Reference) | | 1 (Reference) | |
| G/A [a] | 55 | 15 (28%) | 39 (72%) | | 2.1 (1.8, 2.7) | 1.19 (0.85, 1.66) | | 1.21 (0.87, 1.69) | | 5.4 (3.6, 8.7) | 1.20 (0.85, 1.69) | | 1.18 (0.83, 1.67) | |

(*Continued*)

**Table 2.** (Continued)

| Genotype | N | Tumor response | | | Progression-free survival | | | | | Overall survival | | | | |
|---|---|---|---|---|---|---|---|---|---|---|---|---|---|---|
| | | PR+SD | PD | *P* value* | Median, months (95%CI) | HR (95%CI) † | *P* value* | HR (95%CI) ‡ | *P* value* | Median, months (95%CI) | HR (95%CI) † | *P* value* | HR (95%CI) ‡ | *P* value* |
| A/A [a] | 6 | 4 (80%) | 1 (20%) | | | | | | | | | | | |

Abbreviations: PR, partial response; SD, stable disease; PD, progressive disease; HR, hazard ratio; CI, confidence interval.

* *P* value based on Fisher's exact test for tumor response; log-rank test for progression-free survival (PFS) and overall survival (OS) in the univariate analysis (†); and Wald test for PFS and OS in the multivariable Cox regression model adjusted for time to start of regorafenib treatment (<18 vs. ≥18 months), ECOG performance status (0 vs. 1 or 2), primary tumor resection (yes vs. no), and Kohne score (low-intermediate vs. high) (‡). P values < 0.050 are shown in bold text.

[a] In the dominant model. + Estimates not yet reached.

0.29–0.94, adjusted *P* = 0.031) (Table 3 and Fig 1A); in addition, *SMAD2* rs1792671 showed significant association with PFS in both univariate and multivariable analyses (*P* = 0.025, adjusted *P* = 0.047) (Table 3). Similarly, female patients in the validation cohort with any G allele in *INHBA* rs2237432 showed longer OS which was marginally significant in multivariable analysis (adjusted *P* = 0.059) (Table 4 and Fig 1B). After *P*-act multiple testing, the effects were not significant.

In the discovery cohort, male patients with any G allele in *FOXO3* rs12212067 had a significantly shorter PFS and OS than those with T/T allele in both univariate and multivariable model (PFS: *P* = 0.025, adjusted *P* = 0.009; OS: *P* = 0.015, adjusted *P* = 0.006) (Table 3 and Fig 2A). After *P*-act multiple testing, the effects remained significant for both PFS and OS (*P*-act = 0.035 and 0.024, respectively). In the validation cohort, male patients carrying a *FOXO3* rs12212067 T/G allele had a significant longer OS (*P* = 0.040, adjusted *P* = 0.069) (Table 4 and Fig 2B).

## Associations between cachexia SNPs and outcome in the control cohort

Within the control cohort, no significant associations were observed between the cachexia SNPs and outcome in female patients (Table 5).

## Discussion

Our findings present the first evidence that germline variations in the cancer cachexia pathway are associated with outcome in chemorefractory mCRC patients treated with regorafenib. Furthermore, these associations may depend on gender. We analyzed data from 230 patients receiving regorafenib treatment in two cohorts. Among female patients in the Italian regorafenib discovery cohort, those with any G allele in *INHBA* rs2237432 had significantly better OS than those with an A/A variant. A similar association was confirmed in the Japanese regorafenib validation cohort.

Activin A (*INHBA*), a member of the *TGF-beta* superfamily, is a homodimer formed from two inhibin betaA chains [21] which is produced by several cell types and is involved in several physiologic functions, including embryogenesis, cell growth, differentiation, immune response, and angiogenesis [22]. Activins act via heteromeric complexes of two related transmembrane type I (*ACVR1B*, *C*) and type II (*ACVR2B*) serine/threonine kinase receptors to activate the downstream *SMAD* signaling pathway [13]. Circulating activin A level is associated with cachexia syndrome, and increased concentrations in cancer cachectic patients may contribute to the development of this condition [12]. In addition, a model of activin A overexpression in

**Table 3. Association between cachexia-related gene polymorphism and clinical outcome by gender subgroup in the discovery cohort.**

| Genotype | N | Tumor response | | | Progression-free survival | | | | | Overall survival | | | | |
|---|---|---|---|---|---|---|---|---|---|---|---|---|---|---|
| | | PR+SD | PD | P value* | Median, months (95%CI) | HR (95%CI) † | P value* | HR (95%CI) ‡ | P value* | Median, months (95%CI) | HR (95%CI) † | P value* | HR (95%CI) ‡ | P value* |
| **Female** | | | | | | | | | | | | | | |
| ***INHBA* rs2237432** | | | | 0.79 | | | 0.77 | | 0.59 | | | **0.021** | | **0.031** |
| A/A | 37 | 9 (27%) | 24 (73%) | | 1.8 (1.7, 2.3) | 1 (Reference) | | 1 (Reference) | | 4.3 (2.9, 6.5) | 1 (Reference) | | 1 (Reference) | |
| A/G [a] | 24 | 7 (29%) | 17 (71%) | | 2.0 (1.8, 2.3) | 0.93 (0.57, 1.51) | | 1.16 (0.67, 2.02) | | 7.6 (5.4, 10.2) | 0.57 (0.34, 0.95) | | 0.53 (0.29, 0.94) | |
| G/G [a] | 8 | 1 (13%) | 7 (88%) | | | | | | | | | | | |
| ***MSTN* rs7570532** | | | | 0.82 | | | 0.29 | | 0.59 | | | 0.51 | | 0.74 |
| A/A | 41 | 11 (30%) | 26 (70%) | | 2.1 (1.8, 2.3) | 1 (Reference) | | 1 (Reference) | | 6.0 (4.3, 9.0) | 1 (Reference) | | 1 (Reference) | |
| A/G [a] | 23 | 5 (22%) | 18 (78%) | | 1.8 (1.7, 1.9) | 1.29 (0.79, 2.11) | | 1.17 (0.67, 2.03) | | 4.7 (2.6, 7.6) | 1.18 (0.71, 1.96) | | 1.10 (0.62, 1.97) | |
| G/G [a] | 5 | 1 (20%) | 4 (80%) | | | | | | | | | | | |
| ***SMAD2* rs1792671** | | | | 0.39 | | | **0.025** | | **0.047** | | | 0.98 | | 0.95 |
| G/G | 29 | 5 (18%) | 23 (82%) | | 1.9 (1.7, 2.1) | 1 (Reference) | | 1 (Reference) | | 5.9 (3.7, 9.0) | 1 (Reference) | | 1 (Reference) | |
| G/A [a] | 30 | 9 (33%) | 18 (67%) | | 1.9 (1.8, 3.0) | 0.61 (0.35, 1.03) | | 0.56 (0.32, 0.99) | | 5.6 (3.4, 7.9) | 0.99 (0.60, 1.64) | | 0.98 (0.59, 1.64) | |
| A/A [a] | 10 | 3 (30%) | 7 (70%) | | | | | | | | | | | |
| ***FOXO3* rs12212067** | | | | 0.71 | | | 0.62 | | 0.71 | | | 0.32 | | 0.82 |
| T/T | 56 | 15 (28%) | 39 (72%) | | 1.9 (1.8, 2.1) | 1 (Reference) | | 1 (Reference) | | 6.0 (4.1, 8.0) | 1 (Reference) | | 1 (Reference) | |
| T/G | 13 | 2 (18%) | 9 (82%) | | 1.8 (1.0, 3.0) | 1.17 (0.62, 2.20) | | 0.87 (0.40, 1.86) | | 4.5 (2.6, 5.8) | 1.38 (0.71, 2.70) | | 1.09 (0.51, 2.33) | |
| **Male** | | | | | | | | | | | | | | |
| ***INHBA* rs2237432** | | | | 0.48 | | | 0.93 | | 0.84 | | | 0.43 | | 0.21 |
| A/A | 41 | 18 (45%) | 22 (55%) | | 2.3 (2.0, 3.8) | 1 (Reference) | | 1 (Reference) | | 9.4 (6.4, 12.0) | 1 (Reference) | | 1 (Reference) | |
| A/G [a] | 32 | 11 (35%) | 20 (65%) | | 2.2 (1.8, 3.1) | 0.98 (0.63, 1.53) | | 0.95 (0.60, 1.51) | | 5.4 (3.1, 8.9) | 1.19 (0.75, 1.90) | | 1.37 (0.84, 2.24) | |
| G/G [a] | 8 | 4 (57%) | 3 (43%) | | | | | | | | | | | |
| ***MSTN* rs7570532** | | | | 0.56 | | | 0.87 | | 0.51 | | | 0.62 | | 0.55 |
| A/A | 48 | 22 (47%) | 25 (53%) | | 2.5 (1.9, 3.8) | 1 (Reference) | | 1 (Reference) | | 8.7 (5.6, 10.7) | 1 (Reference) | | 1 (Reference) | |
| A/G [a] | 31 | 10 (34%) | 19 (66%) | | 2.1 (1.8, 2.8) | 0.97 (0.61, 1.52) | | 1.18 (0.72, 1.94) | | 5.3 (3.6, 9.6) | 0.89 (0.56, 1.44) | | 1.17 (0.70, 1.95) | |
| G/G [a] | 2 | 1 (50%) | 1 (50%) | | | | | | | | | | | |
| ***SMAD2* rs1792671** | | | | 0.22 | | | 0.17 | | 0.61 | | | 0.19 | | 0.39 |

*(Continued)*

**Table 3.** (Continued)

| Genotype | N | Tumor response | | | Progression-free survival | | | | | Overall survival | | | | |
|---|---|---|---|---|---|---|---|---|---|---|---|---|---|---|
| | | PR+SD | PD | *P* value* | Median, months (95%CI) | HR (95%CI) † | *P* value* | HR (95%CI) ‡ | *P* value* | Median, months (95%CI) | HR (95%CI) † | *P* value* | HR (95%CI) ‡ | *P* value* |
| G/G | 26 | 13 (52%) | 12 (48%) | | 2.8 (1.9, 3.8) | 1 (Reference) | | 1 (Reference) | | 9.1 (4.4, 11.9) | 1 (Reference) | | 1 (Reference) | |
| G/A | 34 | 15 (45%) | 18 (55%) | | 2.8 (1.8, 3.9) | 1.04 (0.62, 1.75) | | 1.07 (0.63, 1.83) | | 8.3 (5.0, 13.9) | 0.88 (0.51, 1.50) | | 0.85 (0.49, 1.48) | |
| A/A | 20 | 5 (26%) | 14 (74%) | | 1.9 (1.8, 2.3) | 1.63 (0.89, 2.99) | | 1.40 (0.71, 2.76) | | 4.9 (2.3, 10.1) | 1.46 (0.79, 2.68) | | 1.35 (0.70, 2.58) | |
| *FOXO3* rs12212067 | | | | 0.026 | | | 0.025 | | 0.009 | | | 0.015 | | 0.006 |
| T/T | 58 | 27 (48%) | 29 (52%) | | 2.6 (2.1, 3.9) | 1 (Reference) | | 1 (Reference) | | 9.1 (6.3, 10.7) | 1 (Reference) | | 1 (Reference) | |
| T/G [a] | 21 | 4 (20%) | 16 (80%) | | 2.0 (1.6, 2.3) | 1.71 (1.03, 2.84) | | 1.99 (1.19, 3.34) | | 4.3 (2.3, 8.7) | 1.83 (1.08, 3.09) | | 2.17 (1.25, 3.75) | |
| G/G [a] | 1 | 1 (100%) | 0 (0%) | | | | | | | | | | | |

Abbreviations: PR, partial response; SD, stable disease; PD, progressive disease; HR, hazard ratio; CI, confidence interval.

* *P* value based on Fisher's exact test for tumor response; log-rank test for progression-free survival (PFS) and overall survival (OS) in the univariate analysis (†); and Wald test for PFS and OS in the multivariable Cox regression model adjusted for time to start of regorafenib treatment (<18 vs. ≥18 months), ECOG performance status (0 vs. 1 or 2), primary tumor resection (yes vs. no), and Kohne score (low-intermediate vs. high) (‡). P values < 0.050 are shown in bold text.

[a] In the dominant model.

+ Estimates not yet reached.

muscle showed upregulation of muscle *Fn14* during muscle wasting [23]. Furthermore, almost all patients with stage IV CRC (93%) have enhanced tumor expression of activin compared with only 40% of patients with stage I CRC [24]. These data indicate that activin expression is higher in more advanced CRC. *INHBA* rs2237432 is reported to have a significant association with fertility [25], although the clinical significance in cancer remains unknown.

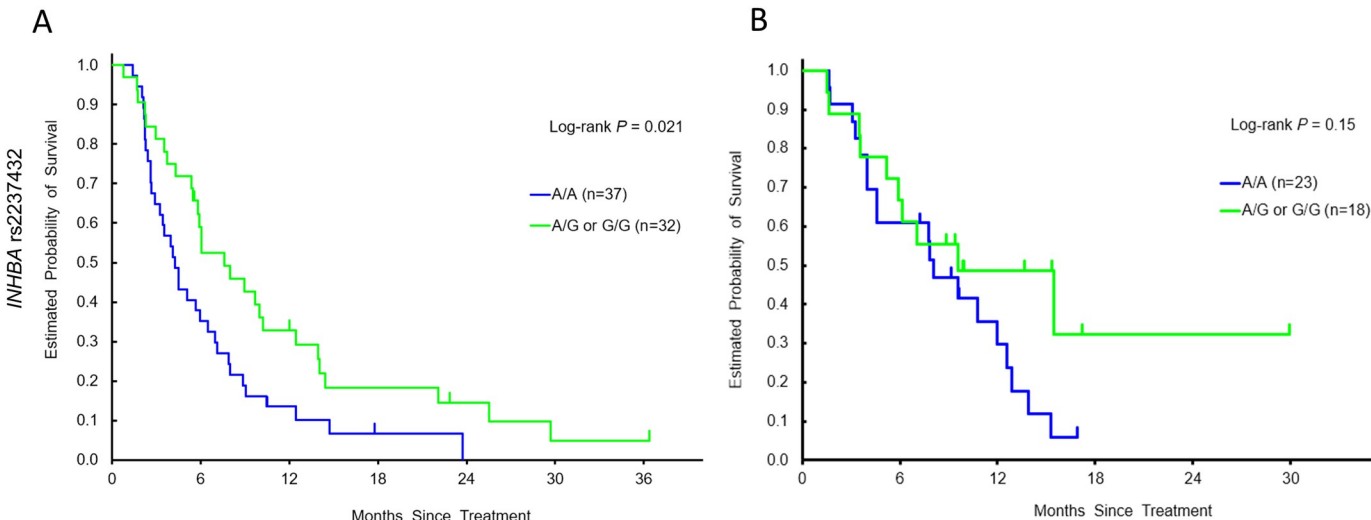

**Fig 1.** Kaplan–Meier cumulative overall survival probability curves stratified by *INHBA* rs2237432 in female patients in (A) the discovery cohort and (B) the validation cohort.

**Table 4. Association between cachexia-related gene polymorphism and clinical outcome by gender subgroup in the validation cohort.**

| Genotype | N | Tumor response | | | Progression-free survival | | | | | Overall survival | | | | |
|---|---|---|---|---|---|---|---|---|---|---|---|---|---|---|
| | | SD | PD | *P* value* | Median, months (95%CI) | HR (95%CI) † | *P* value* | HR (95%CI) ‡ | *P* value* | Median, months (95%CI) | HR (95%CI) † | *P* value* | HR (95%CI) ‡ | *P* value* |
| **Female** | | | | | | | | | | | | | | |
| ***INHBA* rs2237432** | | | | 0.51 | | | 0.55 | | 0.84 | | | 0.15 | | **0.059** |
| A/A | 23 | 7 (39%) | 11 (61%) | | 1.8 (1.8, 3.3) | 1 (Reference) | | 1 (Reference) | | 8.1 (4.0, 12.6) | 1 (Reference) | | 1 (Reference) | |
| A/G [a] | 15 | 5 (50%) | 5 (50%) | | 1.8 (1.1, 2.5) | 1.20 (0.64, 2.25) | | 1.07 (0.55, 2.08) | | 9.6 (5.2, 29.9+) | 0.58 (0.27, 1.25) | | 0.46 (0.21, 1.03) | |
| G/G [a] | 3 | 0 (0%) | 2 (100%) | | | | | | | | | | | |
| ***MSTN* rs7570532** | | | | 0.56 | | | 0.42 | | 0.83 | | | 0.81 | | 0.46 |
| A/A | 22 | 6 (40%) | 9 (60%) | | 1.8 (1.1, 3.3) | 1 (Reference) | | 1 (Reference) | | 9.6 (4.0, 13.9) | 1 (Reference) | | 1 (Reference) | |
| A/G [a] | 18 | 5 (36%) | 9 (64%) | | 2.0 (1.6, 3.3) | 0.79 (0.42, 1.49) | | 0.93 (0.48, 1.82) | | 8.1 (3.6, 12.9) | 1.09 (0.53, 2.28) | | 1.32 (0.62, 2.81) | |
| G/G [a] | 1 | 1 (100%) | 0 (0%) | | | | | | | | | | | |
| ***SMAD2* rs1792671** | | | | 1.00 | | | 0.81 | | 0.68 | | | 0.91 | | 0.68 |
| G/G | 31 | 8 (38%) | 13 (62%) | | 1.8 (1.5, 2.5) | 1 (Reference) | | 1 (Reference) | | 9.6 (4.0, 12.9) | 1 (Reference) | | 1 (Reference) | |
| G/A | 10 | 4 (44%) | 5 (56%) | | 1.9 (0.9, 3.7) | 1.09 (0.51, 2.29) | | 0.84 (0.36, 1.93) | | 7.8 (4.0, 15.3) | 1.05 (0.46, 2.37) | | 0.83 (0.33, 2.04) | |
| ***FOXO3* rs12212067** | | | | 0.61 | | | 0.072 | | 0.30 | | | 0.56 | | 0.83 |
| T/T | 35 | 9 (36%) | 16 (64%) | | 1.8 (1.7, 2.0) | 1 (Reference) | | 1 (Reference) | | 8.1 (5.8, 12.0) | 1 (Reference) | | 1 (Reference) | |
| T/G [a] | 6 | 2 (40%) | 3 (60%) | | 3.3 (0.5, 11.9) | 0.53 (0.21, 1.30) | | 0.58 (0.21, 1.62) | | 13.9 (1.7, 16.9+) | 0.76 (0.28, 2.02) | | 0.90 (0.33, 2.44) | |
| G/G [a] | 1 | 1 (100%) | 0 (0%) | | | | | | | | | | | |
| **Male** | | | | | | | | | | | | | | |
| ***INHBA* rs2237432** | | | | 0.52 | | | 0.43 | | 0.24 | | | 0.73 | | 0.62 |
| A/A | 16 | 8 (53%) | 7 (47%) | | 2.3 (1.7, 4.2) | 1 (Reference) | | 1 (Reference) | | 7.6 (4.1, 27.7+) | 1 (Reference) | | 1 (Reference) | |
| A/G [a] | 14 | 9 (69%) | 4 (31%) | | 2.8 (1.7, 4.6) | 0.77 (0.39, 1.51) | | 0.66 (0.33, 1.33) | | 10.3 (4.0, 27.2+) | 0.86 (0.36, 2.08) | | 1.27 (0.50, 3.23) | |
| G/G [a] | 6 | 2 (40%) | 3 (60%) | | | | | | | | | | | |
| ***MSTN* rs7570532** | | | | 0.45 | | | 0.091 | | 0.61 | | | **0.023** | | 0.13 |
| A/A | 19 | 8 (47%) | 9 (53%) | | 2.0 (1.3, 3.0) | 1 (Reference) | | 1 (Reference) | | 6.3 (4.0, 12.9) | 1 (Reference) | | 1 (Reference) | |
| A/G [a] | 12 | 7 (64%) | 4 (36%) | | 3.3 (2.0, 6.2) | 0.58 (0.29, 1.14) | | 0.83 (0.41, 1.70) | | 26.7+ (6.5, 26.7+) | 0.37 (0.15, 0.92) | | 0.47 (0.18, 1.24) | |
| G/G [a] | 5 | 4 (80%) | 1 (20%) | | | | | | | | | | | |
| ***SMAD2* rs1792671** | | | | 0.35 | | | 0.38 | | 0.19 | | | 0.82 | | 0.18 |

*(Continued)*

**Table 4.** (Continued)

| Genotype | N | Tumor response | | | Progression-free survival | | | | | Overall survival | | | | |
|---|---|---|---|---|---|---|---|---|---|---|---|---|---|---|
| | | SD | PD | *P* value* | Median, months (95%CI) | HR (95%CI) † | *P* value* | HR (95%CI) ‡ | *P* value* | Median, months (95%CI) | HR (95%CI) † | *P* value* | HR (95%CI) ‡ | *P* value* |
| G/G | 26 | 14 (61%) | 9 (39%) | | 2.3 (1.8, 3.2) | 1 (Reference) | | 1 (Reference) | | 11.8 (5.1, 27.7+) | 1 (Reference) | | 1 (Reference) | |
| G/A [a] | 8 | 5 (63%) | 3 (38%) | | 3.8 (1.7, 4.7) | 0.72 (0.33, 1.54) | | 0.58 (0.26, 1.30) | | 8.7 (3.1, 27.2+) | 1.11 (0.43, 2.88) | | 2.09 (0.71, 6.16) | |
| A/A [a] | 2 | 0 (0%) | 2 (100%) | | | | | | | | | | | |
| *FOXO3* rs12212067 | | | | 0.24 | | | 0.18 | | 0.47 | | | **0.040** | | 0.069 |
| T/T | 28 | 12 (48%) | 13 (52%) | | 2.3 (1.7, 3.0) | 1 (Reference) | | 1 (Reference) | | 6.3 (4.0, 10.3) | 1 (Reference) | | 1 (Reference) | |
| T/G | 10 | 7 (78%) | 2 (22%) | | 3.7 (1.9, 7.2) | 0.61 (0.29, 1.29) | | 0.75 (0.34, 1.65) | | 27.2+ (1.9, 27.2+) | 0.30 (0.09, 1.02) | | 0.30 (0.08, 1.10) | |

Abbreviations: SD, stable disease; PD, progressive disease; HR, hazard ratio; CI, confidence interval.

* *P* value based on Fisher's exact test for tumor response, log-rank test for progression-free survival (PFS) and overall survival (OS) in the univariate analysis (†), and Wald test for PFS and OS in the multivariable Cox regression model adjusted for liver metastasis and lymph node involvement (‡). P values < 0.1 are shown in bold text.

[a] In the dominant model.

+ Estimates not yet reached.

Activin has been associated with angiogenesis, but unlike the positive correlation between activin overexpression and cancer cachexia, several studies have reported conflicting data on the relationship of activin overexpression with angiogenesis in various tissue types. Activin A increases *VEGF* expression via the physical interaction of *SMAD2* with the *MAPK*-regulated transcription factor SP1 in hepatocellular carcinoma [26]. In contrast, activin A acts as a tumor suppressor in neuroblastoma [27] and gastric cancer [28] cells via the inhibition of

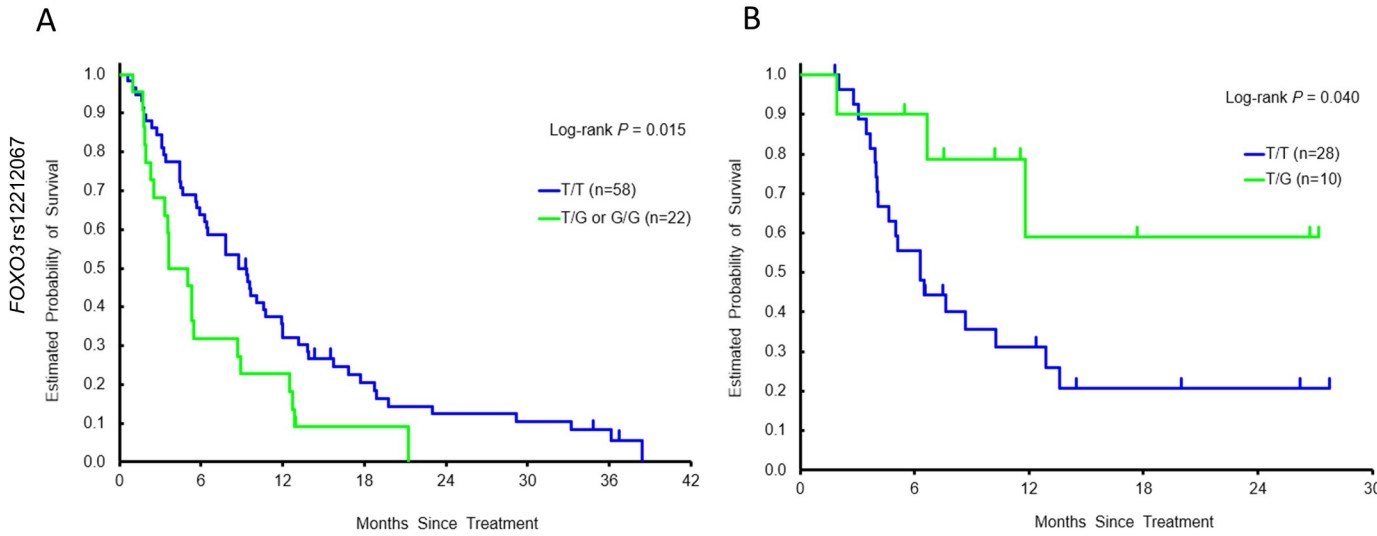

**Fig 2.** Kaplan–Meier cumulative overall survival probability curves stratified by *FOXO3* rs12212067 in male patients in (A) the discovery cohort and (B) the validation cohort.

**Table 5. Association between *INHBA* rs2237432 and clinical outcome in female patients in the control cohort (Italian TAS-102 cohort).**

| Genotype | N | Tumor response | | | Progression-free survival | | | | | Overall survival | | | | |
|---|---|---|---|---|---|---|---|---|---|---|---|---|---|---|
| | | SD | PD | *P* value* | Median, months (95%CI) | HR (95%CI) † | *P* value* | HR (95%CI) ‡ | *P* value* | Median, months (95%CI) | HR (95%CI) † | *P* value* | HR (95%CI) ‡ | *P* value* |
| ***INHBA* rs2237432** | | | | 1.00 | | | 0.28 | | 0.85 | | | 0.097 | | 0.24 |
| A/A | 26 | 7 (27%) | 19 (73%) | | 2.3 (1.9, 2.6) | 1 (Reference) | | 1 (Reference) | | 7.3+ (3.7, 7.3+) | 1 (Reference) | | 1 (Reference) | |
| A/G [a] | 22 | 6 (27%) | 16 (73%) | | 2.0 (1.7, 2.4) | 1.38 (0.73, 2.60) | | 1.07 (0.55, 2.06) | | 4.1 (2.7, 5.5 +) | 2.03 (0.85, 4.86) | | 1.77 (0.69, 4.56) | |
| G/G [a] | 1 | 0 (0%) | 1 (100%) | | | | | | | | | | | |

Abbreviation: SD, stable disease; PD, progressive disease; HR, hazard ratio; CI, confidence interval.

* P value based on Fisher's exact test for response; log-rank test in the univariate analysis (†); and Wald test in the multivariate analysis within Cox regression model adjusted for age group (<61 vs ≥61), liver metastasis, ECOG performance status, previous anti-EGFR therapy (‡). P values < 0.050 are shown in bold text.

[a] In the dominant model.

+ Estimates not yet reached.

*VEGF* mediated-angiogenesis. These findings suggest that activin has dual proinflammatory and anti-inflammatory roles, depending on the cell type and stage of cancer development. Another potential activin-related mechanism is *CCL/CCR*-dependent angiogenesis. *CCL2* binds its receptor *CCR2* to promote angiogenesis by recruiting macrophages [29, 30]. Activin has a critical role in controlling the expression of *CCL2/CCR2* in macrophages by increasing *CCR2* expression while inhibiting *CCL2* expression [31]. Regorafenib is a small molecule that inhibits various intracellular kinases involved in tumor angiogenesis, metastasis, oncogenesis, and tumor immunity. Our results especially found a correlation with tumor angiogenesis. Regorafenib inhibits tumor angiogenesis through inhibiting VEGFR1-3 and TIE2. A preclinical study indicated that *INHBA* exerts diverse effects on the VEGF pathway, including upregulation of the ligand, VEGF, as well as VEGF receptors [16]. Considering these data together, *INHBA* polymorphism may be associated with the effect of regorafenib through exerting its actions via VEGFR.

*INHBA* rs2237432 is an intronic SNP that is classified as a synonymous SNP. Generally, non-synonymous SNPs are considered to affect gene behavior even more considerably than synonymous SNPs. However, some intronic SNPs may affect gene splicing or expression, and such SNPs may have an effect on the function of a gene [32, 33]. Indeed, prediction tools revealed that *INHBA* rs2237432 might have a role as a strong enhancer of *INHBA* expression [33]. These suggest that rs2237432 is associated with the expression of *INHBA*.

In this study, the association between *INHBA* rs2237432 and clinical outcome was demonstrated only in female patients. Activin is an important modulator of follicle-stimulating synthesis and secretion of hormones such as estrogen and progesterone [34, 35]. Several studies have demonstrated the importance of activin and estrogen crosstalk during cancer initiation [36–38]. In addition, estrogen is reported to suppress activin subunit gene promoter activities [39], suggesting that activin activities differ by gender. These results may explain why the association between activin polymorphism and clinical outcome was dependent on gender. Unfortunately, however, because of the lack of samples we were unable to evaluate estrogen levels.

We included 128 patients who were treated with TAS-102 as the control cohort. TAS-102 is an oral drug that combines trifluridine and thymidine phosphorylase inhibitor [40]. The main antitumor effect of TAS-102 is due to DNA dysfunction by trifluridine incorporation into DNA

[41]. In the TAS-102 cohort, contrary to the regorafenib cohort, *INHBA* rs2237432 at any G allele showed a trend toward worse PFS and OS compared with that at the A/A allele. However, these differences did not reach statistical significance, indicating that *INHBA* rs2237432 has a specific association with regorafenib efficacy. A recent retrospective study showed comparable efficacy between regorafenib and TAS-102 [42]. However, a systemic review demonstrated that regorafenib was associated with more toxicity compared with TAS-102 [43]. On the basis of these results, female patients with the *INHBA* rs2237432 A/A allele should avoid regorafenib treatment and be treated with TAS-102 or best supportive care. Such a biomarker-based strategy will identify patients who are eligible for regorafenib treatment, resulting in improved clinical outcomes and quality of life for all patients treated with regorafenib.

Our study also indicated that the impact of *FOXO3* rs12212067 on OS different significantly between the discovery and validation cohorts. This finding may result from etiological differences between Japanese and Italian populations. Several studies have shown that *FOXO3* rs12212067 is associated with the clinical course of inflammatory diseases such as Crohn's disease or rheumatoid arthritis [44, 45]. *FOXO3* has also been linked to the regulation of immune responses using systems biology [46] and knockout mouse models [47]. Furthermore, the *FOXO3* rs12212067 T/T allele is significantly associated with increased inflammatory cytokine production by monocytes (*IL-6*, *IL-8*, *IL-1beta* and *TNF-alfa*) compared with the G/G variant [48]. The pathogenesis of cachexia may be influenced by various factors, including genetic predisposition, inflammatory cytokines, and hormonal aspects. Especially, SNPs within inflammatory cytokine genes can affect cytokine levels and the degree of inflammation, and these SNP functions are reported to differ according to ethnicity [49].

This study had some limitations, such as the sample size and the retrospective design. In addition, we were unable to correlate the *INHBA* and *FOXO3* polymorphisms with intratumoral or serum expression levels, which may have clarified the mechanisms of regorafenib resistance. We were also unable to determine the relationship between the polymorphisms and skeletal muscle mass. In addition, this study presents no information on *RAS* mutation status in the Japanese cohort. We did not confirm our previous results showing that the *ACVR2B* rs2268753 genotype was associated with survival in *RAS* mutant mCRC patients receiving first-line anti-VEGF therapy, which warrants further investigation.

In conclusion, we evaluated for the first time the association of genetic variations in cancer cachexia-associated genes with clinical outcome in mCRC patients treated with regorafenib. We found that *INHBA* rs2237432 was significantly associated with clinical outcomes in female mCRC patients treated with regorafenib. Our findings may contribute to the identification of predictive or prognostic biomarkers of regorafenib therapy and potential drug targets in mCRC patients with cancer cachexia. Further studies are required, however, to fully elucidate the underlying biological mechanisms of the cachexia disease pathway.

## Supporting information

**S1 Table. Polymorphisms and primers.**
(DOCX)

**S2 Table. Association between cachexia-related gene polymorphism and clinical outcome in the validation cohort (Japanese regorafenib cohort).**
(DOCX)

## Acknowledgments

We thank Clare Cox, PhD, and H. Nikki March, PhD, from Edanz Group (www.edanzediting.com/ac) for editing a draft of this manuscript.

## Author Contributions

**Conceptualization:** Yuji Miyamoto, Heinz-Josef Lenz.

**Data curation:** Marta Schirripa, Mitsukuni Suenaga, Shu Cao, Fotios Loupakis.

**Formal analysis:** Shu Cao, Dongyun Yang.

**Investigation:** Yuji Miyamoto, Marta Schirripa, Mitsukuni Suenaga, Satoshi Okazaki, Martin D. Berger, Satoshi Matsusaka.

**Resources:** Marta Schirripa, Mitsukuni Suenaga, Fotios Loupakis, Sara Lonardi, Filippo Pietrantonio, Beatrice Borelli, Chiara Cremolini, Toshiharu Yamaguchi.

**Supervision:** Wu Zhang, Satoshi Okazaki, Martin D. Berger, Yan Ning, Hideo Baba, Heinz-Josef Lenz.

**Writing – original draft:** Yuji Miyamoto.

**Writing – review & editing:** Wu Zhang, Hideo Baba, Heinz-Josef Lenz.

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
