## [Decision Letter · Decision Letter 0]

6 Apr 2020

PONE-D-20-06097

A polymorphism in the cachexia-associated gene INHBA predicts efficacy of regorafenib in patients with refractory metastatic colorectal cancer.

PLOS ONE

Dear Dr Miyamoto,

Thank you for submitting your manuscript to PLOS ONE. After careful consideration, we feel that it has merit but does not fully meet PLOS ONE’s publication criteria as it currently stands. Therefore, we invite you to submit a revised version of the manuscript that addresses the points raised during the review process.

ACADEMIC EDITOR:The study is interesting. Please kindly respond to the reviewers' comments in detail. 

We would appreciate receiving your revised manuscript by May 21 2020 11:59PM. To enhance the reproducibility of your results, we recommend that if applicable you deposit your laboratory protocols in protocols.io, where a protocol can be assigned its own identifier (DOI) such that it can be cited independently in the future. For instructions see: http://journals.plos.org/plosone/s/submission-guidelines#loc-laboratory-protocols

We look forward to receiving your revised manuscript.

Kind regards,

Jason Chia-Hsun Hsieh, M.D. Ph.D

Academic Editor

PLOS ONE

Additional Editor Comments (if provided):

The study is interesting. Please kindly respond to the reviewers' comments in detail.

2. Please provide additional details regarding participant consent. In the ethics statement in the Methods and online submission information, please ensure that you have specified what type of consent you obtained (for instance, written or verbal, and if verbal, how it was documented and witnessed). If your study included minors, state whether you obtained consent from parents or guardians.

3. Thank you for stating that the patients provided informed consent for their blood samples to be used for the purposes of research in this study. However, we also note that patients medical records are also used for statistical analysis in your study.

In the ethics statement in the manuscript and in the online submission form, please provide additional information about the patient records used in your retrospective study, including: a) whether all data were fully anonymized before you accessed them; b) the date range (month and year) during which patients' medical records were accessed; c) the date range (month and year) during which patients whose medical records were selected for this study sought treatment. If patients provided informed written consent to have data from their medical records used in research, please include this information.

4. For studies involving humans categorized by race/ethnicity, authors should update outmoded terms and potentially stigmatizing labels to more current, acceptable terminology. For example, “Caucasian” should be changed to “white” or “of [Western] European descent” (as appropriate).

5. Please provide a sample size and power calculation in the Methods, or discuss the reasons for not performing one before study initiation.

6. Thank you for stating the following in the Competing Interests section:

"HJ Lenz has received honoraria from Merck Serono, Roche, Celgene, Bayer, and Boehringer Ingelheim. HB received honoraria from Chugai Pharma, Bayer, Taiho Pharmaceutical and Merck Serono. The other authors have no conflict of interest."

7. PLOS requires an ORCID iD for the corresponding author in Editorial Manager on papers submitted after December 6th, 2016. Please ensure that you have an ORCID iD and that it is validated in Editorial Manager. To do this, go to ‘Update my Information’ (in the upper left-hand corner of the main menu), and click on the Fetch/Validate link next to the ORCID field. This will take you to the ORCID site and allow you to create a new iD or authenticate a pre-existing iD in Editorial Manager. Please see the following video for instructions on linking an ORCID iD to your Editorial Manager account: https://www.youtube.com/watch?v=_xcclfuvtxQ

Reviewers' comments:

Reviewer's Responses to Questions

5. Review Comments to the Author

Reviewer #1: This is a well written manuscript. The findings are sound, however, i have some minor comments.

1- In page 10 line 14, criteria of high linkage disequilibrium should be defined.

2- In the discussion, the authors should add a paragraph about the non-significant SNPs and their role, relationship to the topic.

3- The authors should correlate their findings to the clinical setting more clearly. They should write about the application of their findings and how will it improve the follow up of the mCRC patients treated with regorafenib.

4- I see that the discussion don't include a relation of the significant findings with the mechanism of action of regorafenib.

Reviewer #2: PONE-D-20-06097

Title: A polymorphism in the cachexia-associated gene INHBA predicts efficacy of regorafenib in patients with refractory metastatic colorectal cancer.

Overview

The authors aimed to evaluate the prognostic significance of cachexia-associated genetic variants in refractory metastatic colorectal cancer (mCRC) patients treated with regorafenib.

Comments:

1. Where in the paper the authors discuss the control cohort of 128 patients receiving TAS-102? Did the authors compare the results of regorafenib to that of control cohort group?

2. One recent report suggests “regorafenib was associated with more toxicity compared with TAS-102” (Abrahao et al; Clin Colorectal Cancer. 2018 Jun;17(2):113-120). How this correlate with the current analysis? Authors should discuss this in detail.

3. Most of the correlations obtained from discovery cohort was opposite to the results obtained from validation cohort? Why the authors think this is happening? Smaller sample size? Different geographical condition? Authors should discuss this in detail.

4. Page 25, authors state “We analyzed data from 233 patients receiving regorafenib treatment in two cohorts”. However according to methods (page 6-7) it should be 230; which is correct?

5. Authors state “estrogen is reported to suppress activin subunit gene promoter activities, suggesting that activin activities differ by gender. These results may explain why the association between activin polymorphism and clinical outcome was dependent on gender”. This is interesting but this should be validated with the correlative analysis by comparing estrogen level in the patient group.

6. Some validation data in patient’s serum or tissue biopsy would strengthen the impact of the study.

Reviewer #3: The manuscript by Miyamoto and colleagues (PONE-D-20-06097) entitled “A polymorphism in the cachexia-associated gene INHBA predicts efficacy of regorafenib in patients with refractory metastatic colorectal cancer” is a retrospective exploratory study that details whether the nucleotide polymorphisms identified in the Discovery Cohort of 150 regorafenib-treated patients from Padova Italy, the regorafenib-treated Validation Cohort of 50 patients from Japan, compared to the 128 patient Control Cohort from Padova and Milano Italy receiving TAS-102 will be predictive for using regorafenib as the primary therapeutic in mCRC and whether these changes lead to AE or SAE toxicity events in the regorafenib-treated cohorts and are influenced by patient gender. In total the group examined 12 SNPs in 8 genes involved in cachexia, which were validated in an 80-patient cohort from Japan who were also treated with regorafenib.

This retrospective study examined 12 SNPs in genes known to be altered in cachexia, first in the Discovery Cohort, then checking whether that same trends could be seen in the validation Cohort, compared to the TAS-102 treated control, non-regorafenib treated cohort from 2 Phase-III RCTs in mCRC patients that were refractory to other standard treatments. In the end, all of these analyses showed that no statistically significant differences between the Discovery group and the validation group were noted, except for with the INHBA SNP where a T to G transition resulted in higher Overall Survival in both groups only when female patients were included, by both univariate and multivariate analyses. One interesting conclusion was observed with the FOXO3 SNP where male patients in the Discovery Cohort showed reduced PFS and OS compared to the Validation Cohort that displayed increased PFS and OS.

Overall, this retrospective study was very well-written and the data was presented in a clear and concise manner. The results manuscript demonstrated a clear gender-based difference in INHBA SNP effects on Overall Survival in the Discovery Cohort. This finding fits with the role of INHBA as a TGFß super-family member including its role in cell growth, angiogenesis, and synthesis of estrogen and progesterone. I found it was particularly interesting that the Discovery and Validation Cohorts displayed a difference in PFS and OS and how they responded to regorafenib treatment in regard to the FOXO3 SNP since the Discovery Cohort is located in Padova Italy while the Validation Cohort is in Japan. Table 5 shows that no correlation exists in It will be of great interest going forward to see what other factors contribute to this difference where one observed an increase in PFS and OS, while the other displayed a decrease. This group seems the ones likely to determine what is involved since they are the only group in the literature who have examined the role of particular cytokine genetic variants as they did in Suenaga 2018 Clin. Colorectal Cancer 17(2)e395-414 looking at SNPs in CCL3 and CCL4 using the same Discovery and Validation Cohorts used in this study.

Additionally, the authors detailed that there was genotype concordance of >99% when they randomly surveyed 10% of all patients by direct sequencing. Moreover, these investigators were blinded to all the clinical data, helping to minimize/manage or eliminate bias. The authors also mention that there were 4 limitations to their study including (1) sample size, (2) the retrospective nature of the study, and (3) unable to correlate SNPs with intra-tumoral or serum expression levels and (4) the effect of these SNPs on skeletal mass. Even with these limitations, this study still represents another important step forward in better understanding how patient SNP genetics came be used to select the proper treatment regimen as a form of pharmaco-genetics. The authors also include many of the important references in the field including other clinical trials, however they may wish to consider Abrahao et al., 2018 Clin. Colorectal Cancer 17(2):e113-20; Clarke et al., 2019 Cancer Chemother Pharmacol. 84(4):909-17; Fan et al., 2016 Oncotarget 7(39):64136-47; Ricci et al., 2020 World J. Gastrointest. Oncol. 12(3):301-310) that have examined regorafenib and various cytokines like TGFß, TNF�, VEGF, CCL2, CCL5 and CCL4. It would also be of benefit to know if treatment with regorafenib showed any effect in reducing angiogenesis of the patients tumors as one might expect that changes in PFS and OS might be due to blocking the continued or further angiogenic nature of mCRC.

6. PLOS authors have the option to publish the peer review history of their article (what does this mean?). If published, this will include your full peer review and any attached files.

Reviewer #1: No

Reviewer #2: No

Reviewer #3: No

---

## [Author Response · Author response to Decision Letter 0]

24 Jul 2020

RESPONSES TO COMMENTS

We would like to thank the reviewers for their insightful comments, which have helped us to improve our paper.

Reviewer 1

Comment 1: In page 10 line 14, criteria of high linkage disequilibrium should be defined.

Response 1: Thank you for highlighting this. Accordingly, we have added the following text to the Materials and Methods section of the revised manuscript: 

“High linkage disequilibrium was defined as r2 > 0.7.”

Comment 2: In the discussion, the authors should add a paragraph about the non-significant SNPs and their role, relationship to the topic.

Response 2: In accordance with the reviewer’s comment, we have added the following text to the Discussion section of the revised manuscript:

“INHBA rs2237432 is an intronic SNP that is classified as a synonymous SNP. Generally, non-synonymous SNPs are considered to affect gene behavior even more considerably than synonymous SNPs. However, some intronic SNPs may affect gene splicing or expression, and such SNPs may have an effect on the function of a gene [32,33]. Indeed, prediction tools revealed that INHBA rs2237432 might have a role as a strong enhancer of INHBA expression [33]. These suggest that rs2237432 is associated with the expression of INHBA.”

Comment 3: The authors should correlate their findings to the clinical setting more clearly. They should write about the application of their findings and how will it improve the follow up of the mCRC patients treated with regorafenib.

Response 3: Thank you very much for your comments. To address this, we have added the following text to the Discussion:

“A recent retrospective study showed comparable efficacy between regorafenib and TAS-102 [42]. However, a systemic review demonstrated that regorafenib was associated with more toxicity compared with TAS-102 [43]. On the basis of these results, female patients with the INHBA rs2237432 A/A allele should avoid regorafenib treatment and be treated with TAS-102 or best supportive care. Such a biomarker-based strategy will identify patients who are eligible for regorafenib treatment, resulting in improved clinical outcomes and quality of life for all patients treated with regorafenib.”

Comment 4: I see that the discussion don't include a relation of the significant findings with the mechanism of action of regorafenib.

Response 4: Thank you for pointing this out. In accordance with this comment, we have added the following text to the Discussion:

“Regorafenib is a small molecule that inhibits various intracellular kinases involved in tumor angiogenesis, metastasis, oncogenesis, and tumor immunity. Our results especially found a correlation with tumor angiogenesis. Regorafenib inhibits tumor angiogenesis through inhibiting VEGFR1-3 and TIE2. A preclinical study indicated that INHBA exerts diverse effects on the VEGF pathway, including upregulation of the ligand, VEGF, as well as VEGF receptors [16]. Considering these data together, INHBA polymorphism may be associated with the effect of regorafenib through exerting its actions via VEGFR.”

Reviewer 2

Comment 1: Where in the paper the authors discuss the control cohort of 128 patients receiving TAS-102? Did the authors compare the results of regorafenib to that of control cohort group?

Response 1: Apologies for our oversight. In accordance with the reviewer’s comment, we have added the following text to the Discussion: 

“We included 128 patients who were treated with TAS-102 as the control cohort. TAS-102 is an oral drug that combines trifluridine and thymidine phosphorylase inhibitor [40]. The main antitumor effect of TAS-102 is due to DNA dysfunction by trifluridine incorporation into DNA [41]. In the TAS-102 cohort, contrary to the regorafenib cohort, INHBA rs2237432 at any G allele showed a trend toward worse PFS and OS compared with that at the A/A allele. However, these differences did not reach statistical significance, indicating that INHBA rs2237432 has a specific association with regorafenib efficacy. ”

Comment 2: One recent report suggests “regorafenib was associated with more toxicity compared with TAS-102” (Abrahao et al; Clin Colorectal Cancer. 2018 Jun;17(2):113-120). How this correlate with the current analysis? Authors should discuss this in detail.

Response 2: In accordance with the reviewer’s comment, we have added the following text to the Discussion: 

“A recent retrospective study showed comparable efficacy between regorafenib and TAS-102 [42]. However, a systemic review demonstrated that regorafenib was associated with more toxicity compared with TAS-102 [43]. On the basis of these results, female patients with the INHBA rs2237432 A/A allele should avoid regorafenib treatment and be treated with TAS-102 or best supportive care. ”

Comment 3: Most of the correlations obtained from discovery cohort was opposite to the results obtained from validation cohort? Why the authors think this is happening? Smaller sample size? Different geographical condition? Authors should discuss this in detail.

Response 3: Thank you for your comment. We stated in the manuscript: “This finding may result from etiological differences between Japanese and Italian populations.” We have added the following text to the Discussion of the revised manuscript:

“The pathogenesis of cachexia may be influenced by various factors, including genetic predisposition, inflammatory cytokines, and hormonal aspects. Especially, SNPs within inflammatory cytokine genes can affect cytokine levels and the degree of inflammation, and these SNP functions are reported to differ according to ethnicity [49]”

Comment 4: Page 25, authors state “We analyzed data from 233 patients receiving regorafenib treatment in two cohorts”. However according to methods (page 6-7) it should be 230; which is correct?

Response 4: Thank you for highlighting our mistake. We have corrected the patient number to 230 in the revised manuscript.

Comment 5: Authors state “estrogen is reported to suppress activin subunit gene promoter activities, suggesting that activin activities differ by gender. These results may explain why the association between activin polymorphism and clinical outcome was dependent on gender”. This is interesting but this should be validated with the correlative analysis by comparing estrogen level in the patient group.

Response 5: We agree that additional information on serum or tissue biopsy would be valuable. Regrettably, however, because of the lack of samples, we are unable to undertake this experiment. We have added the following text to the Discussion section:

“Unfortunately, however, because of the lack of samples, we were unable to evaluate estrogen levels.”

Comment 6: Some validation data in patient’s serum or tissue biopsy would strengthen the impact of the study.

Response 6: Thank you for your suggestion. However, because of the lack of samples, we are unable to perform this experiment. 

Reviewer 3

Comment 1: The authors also include many of the important references in the field including other clinical trials, however however they may wish to consider Abrahao et al., 2018 Clin. Colorectal Cancer 17(2):e113-20; Clarke et al., 2019 Cancer Chemother Pharmacol. 84(4):909-17; Fan et al., 2016 Oncotarget 7(39):64136-47; Ricci et al., 2020 World J. Gastrointest. Oncol. 12(3):301-310) that have examined regorafenib and various cytokines like TGFß, TNF�, VEGF, CCL2, CCL5 and CCL4.

Response 1: Thank you for your suggestion. We have added a citation in accordance with your suggestion.

43. Abrahao ABK, Ko YJ, Berry S, Chan KKW. A Comparison of Regorafenib and TAS-102 for Metastatic Colorectal Cancer: A Systematic Review and Network Meta-analysis. Clin Colorectal Cancer. 2018;17: 113–120. doi:10.1016/j.clcc.2017.10.016

---

## [Decision Letter · Decision Letter 1]

7 Sep 2020

A polymorphism in the cachexia-associated gene INHBA predicts efficacy of regorafenib in patients with refractory metastatic colorectal cancer.

PONE-D-20-06097R1

Dear Dr. Miyamoto,

We’re pleased to inform you that your manuscript has been judged scientifically suitable for publication and will be formally accepted for publication once it meets all outstanding technical requirements.

Kind regards,

Jason Chia-Hsun Hsieh, M.D. Ph.D

Academic Editor

PLOS ONE

Additional Editor Comments (optional):

I think most of the questions were answered adequately.

Reviewers' comments:

Reviewer's Responses to Questions

**Comments to the Author**

1. If the authors have adequately addressed your comments raised in a previous round of review and you feel that this manuscript is now acceptable for publication, you may indicate that here to bypass the “Comments to the Author” section, enter your conflict of interest statement in the “Confidential to Editor” section, and submit your "Accept" recommendation.

Reviewer #2: All comments have been addressed

2. Is the manuscript technically sound, and do the data support the conclusions?

Reviewer #2: Partly

3. Has the statistical analysis been performed appropriately and rigorously? 

Reviewer #2: Yes

4. Have the authors made all data underlying the findings in their manuscript fully available?

Reviewer #2: (No Response)

5. Is the manuscript presented in an intelligible fashion and written in standard English?

Reviewer #2: (No Response)

6. Review Comments to the Author

Reviewer #2: (No Response)

7. PLOS authors have the option to publish the peer review history of their article (what does this mean?). If published, this will include your full peer review and any attached files.

Reviewer #2: No

---

## [Editor Report · Acceptance letter]

14 Sep 2020

PONE-D-20-06097R1 

A polymorphism in the cachexia-associated gene *INHBA* predicts efficacy of regorafenib in patients with refractory metastatic colorectal cancer 

Dear Dr. Miyamoto:

I'm pleased to inform you that your manuscript has been deemed suitable for publication in PLOS ONE. Congratulations! Your manuscript is now with our production department. 

Kind regards, 

on behalf of

Dr. Jason Chia-Hsun Hsieh 

Academic Editor

PLOS ONE